# Consistent Experience Replay in High-Dimensional Continuous Control with Decayed Hindsights

Xiaoyun Feng 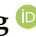

Department of Electronic Engineering and Information Science, University of Science and Technology of China, Hefei 230027, China; xy2012@mail.ustc.edu.cn; Tel.: +86-1829-796-6578

**Abstract:** The manipulation of complex robotics, which is in general high-dimensional continuous control without an accurate dynamic model, summons studies and applications of reinforcement learning (RL) algorithms. Typically, RL learns with the objective of maximizing the accumulated rewards from interactions with the environment. In reality, external rewards are not trivial, which depend on either expert knowledge or domain priors. Recent advances on hindsight experience replay (HER) instead enable a robot to learn from the automatically generated sparse and binary rewards, indicating whether it reaches the desired goals or pseudo goals. However, HER inevitably introduces hindsight bias that skews the optimal control since the replays against the achieved pseudo goals may often differ from the exploration of the desired goals. To tackle the problem, we analyze the skewed objective and induce the decayed hindsight (DH), which enables consistent multi-goal experience replay via countering the bias between exploration and hindsight replay. We implement DH for goal-conditioned RL both in online and offline settings. Experiments on online robotic control tasks demonstrate that DH achieves the best average performance and is competitive with state-of-the-art replay strategies. Experiments on offline robotic control tasks show that DH substantially improves the ability to extract near-optimal policies from offline datasets.

**Keywords:** robotic control; goal-conditioned reinforcement learning; offline reinforcement learning; sparse rewards; experience replay; hindsight bias

## 1. Introduction

Learning and planning without an accurate dynamic model provide a great challenge for complex robotics control. In awareness of the daunting difficulty of physical control, reinforcement learning (RL) [1], which instead treats the environment dynamic as a black box, implicitly learns optimal behaviors from interactions with the environment in pursuit of the maximal accumulated rewards. Combined with deep neural networks, deep RL has achieved great success in both simulated and real-world robotic control tasks, such as dexterous manipulation [2–6], motion control [7–10], navigation [11–13] and so on [14–16]. In the ideal case, the reward function for each task should be an indicator for task solving [2,6]. Nevertheless, most algorithms rely on a delicately designed and appropriately shaped task-specific reward function that reflects the domain knowledge [17,18]. For example, Ref. [6] takes the distance from the gripper to the handle as well as the difference between the current and desired door pose into the reward function for a door-opening task. However, it is impractical to engineer rewards for all the tasks, especially for high-dimensional continuous control tasks with enormous state–action spaces (as shown in Figure 1). As it grows tougher and requires considerably more exploration, the ability to efficiently robotic control with unshaped task-solving rewards is indispensable.

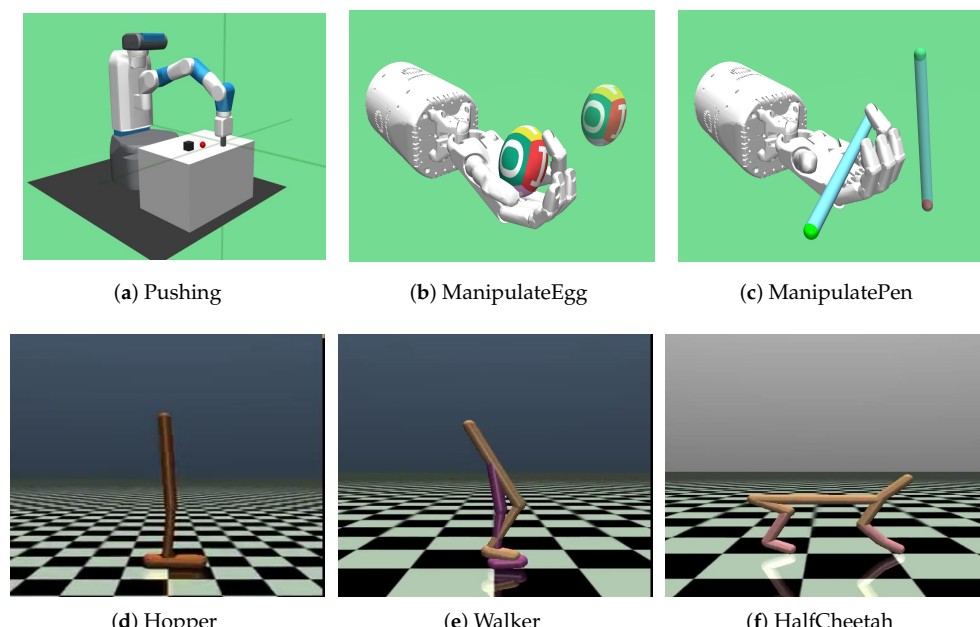

(**a**) Pushing      (**b**) ManipulateEgg      (**c**) ManipulatePen

(**d**) Hopper      (**e**) Walker      (**f**) HalfCheetah

**Figure 1.** High-dimensional continuous control tasks suffering from sparse rewards. (**a**–**c**) Object manipulation tasks with a fetch robotic arm or a shadow dexterous hand; (**d**–**f**) motion control tasks with simulated physics. We consider it a success if an agent achieves the desired goal state within some task-specific tolerance.

The recent advancement on hindsight experience replay (HER) [19] proposes to replay past experiences with pseudo goals (abstracted from states indicating task solving), which enriches pseudo task-solving signals and enables learning from failures. Incorporated with HER, goal-conditioned RL [20–22], where agents pursue various goals in an open-ended environment, learns to achieve and generalize across different goals with sparse task-solving rewards. Recent advances in experience replay with hindsights have shown to greatly enable goal-conditioned RL in various scenarios [23–30]. However, when replaying the experience, it introduces a well-known cognitive bias that describes the overestimation of one's foresights after acquiring the outcomes of executing certain actions, which is called *hindsight bias* [31] and impedes learning in sequential decision-making tasks [32,33]. Concretely, pseudo rewards from the relabeled experience may give the agent unreasonable confidence in reaching past achieved goals, neglecting the actual outcomes of its policy. For value-based learning, the bias leads to overestimation in value function.

In countering the bias of regenerating past experiences with original goals and hindsight goals, existing work either increases the likelihood of sampled actions by assigning aggressive rewards for relabeled experiences [34], or reduces changes in the likelihood of sampled trajectories [35]. However, hindsight replay changes the distribution of replayed goals, yet most of the goals will not be further explored. Thus, assigning excessive hindsight rewards results in the unreliable generalization for goal-conditioned RL, which assumes that behaviors for one goal can generalize to another similar one. As hindsight goals progressively spread across all the goal space in an unprincipled manner, the value function approximation may collapse. Therefore, it is of great significance to restrict excessive hindsights on goals with high uncertainties under the current policy.

Dealing with the mentioned problem, we analyze the biased value function and consider stabilizing it with decayed hindsight rewards, namely decayed hindsight (DH). Compared to vanilla HER, if the current policy cannot regenerate the past experience with a hindsight goal very well, the hindsight reward assigned to the experience will be decreased. Embodying DH into HER enables consistent experience replay, Decayed-HER. For one thing, the inevitable hindsight bias will be adaptively reduced, according to the distribution dissimilarity with replayed goals. For another, it maintains a balance between hindsight

replay and task-solving indication. Both in online and offline settings, we evaluate our proposed Decayed-HER on various goal-conditioned manipulation tasks using a simulated physics engine [36]. Experiments demonstrate that it achieves the best average performance over the state-of-the-art multi-goal replay strategies in online learning and improves the performance in offline learning.

Our contribution can be summarized as below:

- As far as we know, this manuscript is the first to propose a surrogate objective over hindsight experiences after analyzing the hindsight bias in HER, which is the foundation of reliable and unbiased multi-goal value estimation.
- We propose a tractable implementation of the decayed hindsight and successfully embed it to online multi-goal replay and offline RL. Compared to existing works, the proposed method is effective and easy to implement with the potential of getting rid of low sample efficiency and heavy computation burden.
- We conduct a series of experiments and show promising results on high-dimensional continuous control tasks with sparse rewards. Especially for offline RL, the proposed method pictures a prospective future to effectively learn from experiences and datasets without exploring the environment.

## 2. Related Works

*Rewards shaping.* Rewards shaping [17,18] mostly aims at managing every-step reward that heuristically reflects domain knowledge, such as the likelihood of achieving the desired goal. However, as mentioned in [19], HER with shaped rewards [17] performs worse than with sparse rewards. The source of it is that dense shaped rewards are more likely to collapse the value function approximation. We emphasize that manipulation hindsight rewards to counter bias are fundamentally different from reward shaping. In general, bias countering cares about the likelihood of regenerating past experiences by the current policy instead of providing every-step rewards.

Except for task-solving rewards, recent works construct various signals to encourage exploration. Ref. [37] proposesself-balancing shaped rewards, which award sibling rollout to reconstruct a self-balancing optimum. Ref. [38] appliesan asymmetric reward relabeling strategy to induce competition between a pair of agents. Both [37,38] encourage exploration beyond the original tasks.

*Self-Imitation.* Self-imitation [39] shares the similar idea of HER and can be extended to goal-conditioned RL [40–43]. These works imitate achieved trajectories with goal-conditioned policies, just like replaying past experiences with the achieved goals in HER. For self-imitation, the hindsight bias may also lead to unstable policy updates. We will explore it in future work.

## 3. Materials and Methods

In this section, we revise the value-based goal-conditioned RL framework and its universal value function approximators, then analyze experience replay with the hindsight bias mentioned before, and derive Decayed-HER in the following.

### 3.1. Goal-Conditioned RL and Universal Value Function Approximators

Goal-conditioned RL extends typical RL to goal-conditioned behaviors. Consider a discounted Markov decision process (MDP), $(\mathcal{S}, \mathcal{A}, \mathcal{G}, P, r, \gamma)$, where $\mathcal{S}, \mathcal{A}, \mathcal{G}$ each represent a set of states, actions and goals, $P : \mathcal{S} \times \mathcal{A} \times \mathcal{S} \to \mathbb{R}$ and $r : \mathcal{S} \times \mathcal{A} \times \mathcal{G} \to \mathbb{R}$ are the transition probability distribution and reward function, $\gamma \in (0, 1)$ is a discount factor. At the beginning of each episode, the RL agent obtains the desired goal $g \in \mathcal{G}$. Then at time step $t$, it observes a state $s_t \in \mathcal{S}$, executes an action $a_t \in \mathcal{A}$ and receives a reward $r(s_t, a_t, g)$ at the next time step that indicates task solving. Concretely,

$$r_t = r(s_t, a_t, g) = \begin{cases} 0, ||\phi(s_{t+1}) - g||_2 \leq \delta_g \\ -1, \quad otherwise \end{cases}$$

where $\phi : \mathcal{S} \to \mathcal{G}$ defines a tractable mapping from states to goals, and $\delta_g$ is a pre-defined task-specific tolerance threshold [22].

Let $\tau = s_0, a_0, s_1, a_1, \ldots, s_{T-1}, a_{T-1}, s_T$ denote a trajectory, and $R_t = \sum_{i=t}^{T-1} \gamma^{i-t} r_i$ denote its accumulated discounted return at $t \in [0, T-1]$. Let $\pi : \mathcal{S} \times \mathcal{G} \to \mathcal{A}$ denote a universal policy, $V^\pi : \mathcal{S} \times \mathcal{G} \to \mathbb{R}$ and $Q^\pi : \mathcal{S} \times \mathcal{G} \times \mathcal{A} \to \mathbb{R}$ denote its universal value function. Combined with neural networks, universal value function approximators (UVFA) [21] have the expectation of generalizing the value estimation over various goals. With experience $(s, a, s', g, r)$, it optimizes $\pi$ via performing policy improvement with the expectation

$$Q^\pi(s, g, a) := \mathbb{E}_{s'}\left[ r + \gamma \max_{a'} Q^\pi(s', g, a') \right]. \tag{1}$$

In general, HER variants concatenate states and goals together as joint inputs for the $Q$-value function, just like [19,23,44,45].

*3.2. Hindsight Bias*

Dealing with sparse task-solving rewards, HER utilizes the hindsight—the pseudo goal is going to be achieved in the near future—to learn from failures. Concretely, it relabels experience $e = (s, a, s', g, r)$ with a pseudo goal $g_h$ and enables to learn from the relabeled experience $(s, a, s', g_h, r(s, a, g_h))$ with a hindsight reward $r(s, a, g_h)$. Since the pseudo goal is randomly sampled from the same trajectory of the original experience [19], the hindsight reward thus yields a higher probability of goal reaching than $r$. Formally, the current policy $\pi$ is updated with Equation (1) via

$$Q^\pi(s, g_h, a) \leftarrow r(s, a, g_h) + \gamma \max_{a'} Q^\pi(s', g_h, a'),$$

which plays a key role to remember and reuse multi-goal experiences.

However, HER neglects the fact that without loss of generality, we have

$$\pi(s, g) \neq \pi(s, g_h)$$

for most of the states and policies. It introduces hindsight bias indicating that HER inevitably changes the sample distribution of experiences. For experience $e$, we regard the original distribution of experience as $f(e)$, and the distribution of hindsight experience as $h(e)$. The vanilla HER proposes hindsight goals in a random manner, which makes an unpredictable distribution shift from $f(e)$ to $h(e)$ and skews the value estimation over the distribution from $\mathbb{E}_{e \sim f}[Q^\pi(e)]$ to $\mathbb{E}_{e \sim h}[Q^\pi(e)]$. Notice that

$$\mathbb{E}_{e \sim f}[Q^\pi(e)] = \int_e Q^\pi(e) f(e) de = \int_e [\frac{f(e)}{h(e)} Q^\pi(e)] h(e) de$$

$$= \mathbb{E}_{e \sim h}[\frac{f(e)}{h(e)} Q^\pi(e)]. \tag{2}$$

Importance sampling seems natural for canceling the sampling bias but is non-trivial in practice, which may result in policy updates with high variance [22,46].

As the environment dynamic is independent of goals, a goal-conditioned policy is able to regenerate the transition $(s, a, s')$ of $e$ with any input goal as long as, with some probability, it can generate the action $a$ at the state $s$. Without loss of generality, replaying experience with hindsights equals adding diverse behavior policies in advance. Thus we can assume that $f(e) < h(e)$ for most of sampled hindsight experiences but $f(e)$ progressively approaches $h(e)$, as the pseudo goal approaches the original goal. Informally, we assume that

$$\frac{f(e)}{h(e)} < c \leq 1, \quad \Delta Q^\pi(e) = (1 - \frac{f(e)}{h(e)}) Q^\pi(e) > (1-c) Q^\pi(e)$$

for almost every replayed $e$ and some constant $c$.

In order to verify that this theoretical overestimation does exist in practice, we plot the true return and the value estimation of TD3 [47], which already addresses function approximation error in actor-critic methods, on a goal-conditioned continuous control task named parking-v0 [48], in which the ego-vehicle must park in a given space with the appropriate heading. For every epoch during training, we obtain the values with the current policy. Concretely, we randomly sample a $(s, a, g)$ pair from the replay buffer and take its $Q^\pi$-value as value estimation under the current policy $\pi$. Then we take the true discounted returns over 100 rollouts that start with the pair and follow $\pi$ as the return. In Figure 2a, we plot the averaged value estimations and returns over 50 pairs at each epoch. We can see that there is a very clear overestimation during the training process, which may be minimal but cannot be neglected. For one thing, the skewed value function leads to sub-optimal policies. For another, the overestimation may result in a more significant bias for unexplored goals, given the iterated Bellman updates. Countering the effect of hindsight bias is essential.

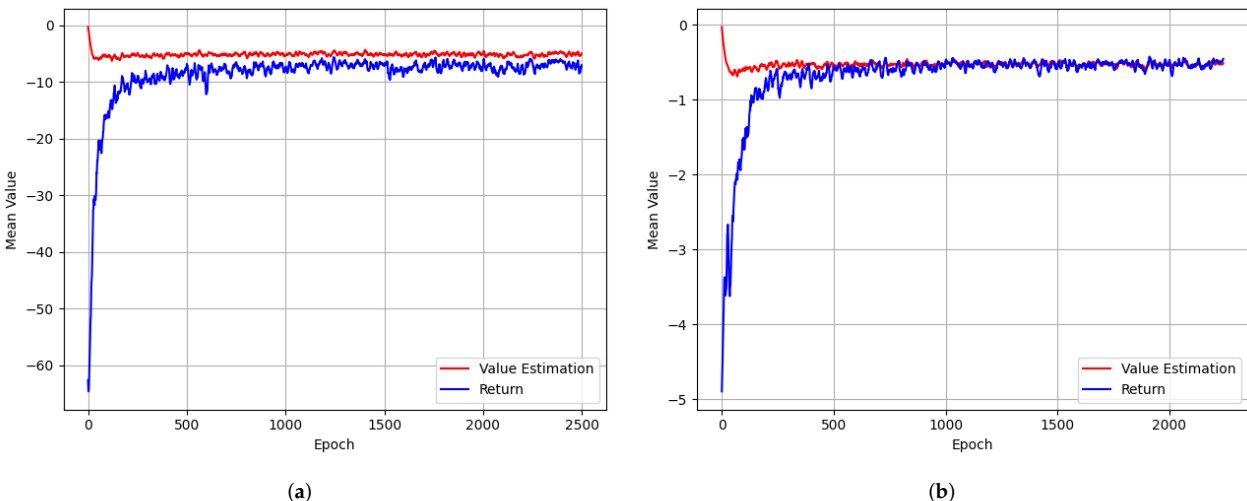

(**a**)                    (**b**)

**Figure 2.** Demonstrating overestimation in the value estimation of TD3 with and without decayed hindsights on parking-v0 during the training process (for approximately 1 million steps). (**a**) TD3 without decayed hindsights; (**b**) TD3 with decayed hindsights.

*3.3. Decayed Hindsights*

In view of Equation (2), we consider the skewed objective $\eta^\pi = \frac{f(e)}{h(e)} Q^\pi(e)$ for relabeled experience $e = (s, a, s', g_h)$. Noticing that importance sampling is non-trivial for hindsight experience, we instead directly learn to approximate $\eta^\pi$ in the derived MDP

$$\eta^\pi(e) = \widetilde{r} + \gamma \mathbb{E}_{e' \sim h}[\eta^\pi(e')]$$

with reward

$$\widetilde{r} = m(e)r - \gamma(1 - m(e))\mathbb{E}_{e' \sim h}[\eta^\pi(e')],$$

where $m(e) = \frac{f(e)}{h(e)}$ describes the distribution dissimilarity and we assume that $h(e) > 0$ holds for every sampled $e$.

**Theorem 1.** *Under mild conditions, $\eta^\pi$ is a surrogate estimate for $Q^\pi$.*

**Proof.** By appending $\widetilde{r} = m(e)r - \gamma(1 - m(e))\mathbb{E}_{e' \sim h}[\eta^\pi(e')]$ into the derived MDP, we can trivially obtain

$$
\begin{aligned}
\mathbb{E}_{e \sim h}[\eta^\pi(e)] &= \mathbb{E}_{e \sim h}[\widetilde{r} + \gamma \mathbb{E}_{e' \sim h}[\eta^\pi(e')]] \\
&= \mathbb{E}_{e \sim h}[m(e)r - \gamma(1 - m(e))\mathbb{E}_{e' \sim h}[\eta^\pi(e')] + \gamma \mathbb{E}_{e' \sim h}[\eta^\pi(e')]] \\
&= \mathbb{E}_{e \sim h}[m(e)r + \gamma m(e)\mathbb{E}_{e' \sim h}[\eta^\pi(e')]] \\
&= \mathbb{E}_{e \sim h}[m(e)(r + \gamma \mathbb{E}_{e' \sim h}[\eta^\pi(e')])] \\
&= \mathbb{E}_{e \sim h}[m(e)(r + \gamma \mathbb{E}_{e' \sim f}[Q^\pi(e')])] \\
&= \mathbb{E}_{e \sim f}[Q^\pi(e)].
\end{aligned}
$$

which indicates that we can approximately estimate the true value $Q^\pi$ over $f$ by self-bootstrapped estimating $\eta^\pi$ over $h$ with $\widetilde{r}$. $\square$

The new MDP inherits the convergence from the original MDP as long as the reward is bounded, i.e., for any experience $e, e'$, bounded $r$ and some constant $C > 0$, we hold $|\widetilde{r}| < C$. It is evident that if $m(e)$ is bounded, the reward is bounded. According to the previous analysis, we can assume that with probability $1 - \delta$, $m(e) \leq 1$ holds. Then it converges and

$$
r - \widetilde{r} = (1 - m(e))(1 + \gamma \mathbb{E}_{e' \sim h}[\eta^\pi(e')]) = (1 - m(e))(1 + \gamma \mathbb{E}_{e' \sim f}[Q^\pi(e')])
$$

is in principle positive for goal-reaching hindsight experience and optimal policy.

On the basis of this observation, we make the following statements:

1. As the difference $r - \widetilde{r}$ grows with $1 - m(e)$, the hindsight reward is in principle decreased with the distribution dissimilarity. We call the reconstructed hindsight rewards *decayed hindsights*.
2. Nevertheless, obtaining $\widetilde{r}$ is non-trivial, as the estimation of $m(e_h)$ can be intractable for hindsight experience $e_h = (s, a, s', g_h, r_h)$. As most of the pseudo goals will not be further explored, we instead consider the ability of the current policy $\pi$ to reproduce the sampled trajectory containing transition $(s, a, s')$, given the original goal $g_o$ and the hindsight goal $g_h$. Under mild conditions, we have

$$
m(e_h)|_{e_h \sim h} = \frac{f(e_h)}{h(e_h)}\Big|_{e_h \sim h} \approx \frac{\mathbb{P}(\tau, g_h)}{\mathbb{P}(\tau, g_o)} \propto \frac{\Pi_i \pi(a_i | s_i, g_h)}{\Pi_i \pi(a_i | s_i, g_o)},
$$

where $\tau = \{s_1, a_1, s_2, a_2, \dots\}$ is the sampled trajectory, and both the probability of sample $g_h$ with $f$ and the probability of sample $g_o$ with $h$ are omitted. (For simplicity, we denote $\mathbb{P}(a = \pi(s, g))$ as $\pi(a|s, g)$.) Concretely, we take

$$
\log m(e)|_{e_h \sim h} \approx \log \frac{\Pi_i \pi(a_i | s_i, g_h)}{\Pi_i \pi(a_i | s_i, g_o)} = \sum_i \log \pi(a_i | s_i, g_h) - \sum_i \log \pi(a_i | s_i, g_o)
$$

$$
m(e_h) \approx exp\{\sum_i (\log \pi(a_i | s_i, g_h) - \log \pi(a_i | s_i, g_o))\}.
$$

For fear of high variance, we constrain $m(e) \in (0, c], c < 1$ in the implementation. In future work, we may pay attention to further easing the computation burden.

After obtaining $m(e)$ and appending

$$
\widetilde{r} = m(e)r(e) - \gamma(1 - m(e))\hat{\eta}^\pi(e')
$$

in the value updates, we modify the vanilla algorithm in solving parking-v0 to TD3 with DH. In Figure 2b, we also plot the averaged value estimations and returns over 50 pairs at each epoch. We can see that the overestimation progressively disappears. As the learning becomes stable, the value estimations match well with the returns. Compared to Figure 2a,

the returns are higher than those obtained by TD3 without decayed hindsights. Figure 2 shows that the hindsight bias does exist and can be decreased by our proposed decayed hindsights.

### 3.4. Decayed-HER

Figure 3 presents the overall framework for Decayed-HER.

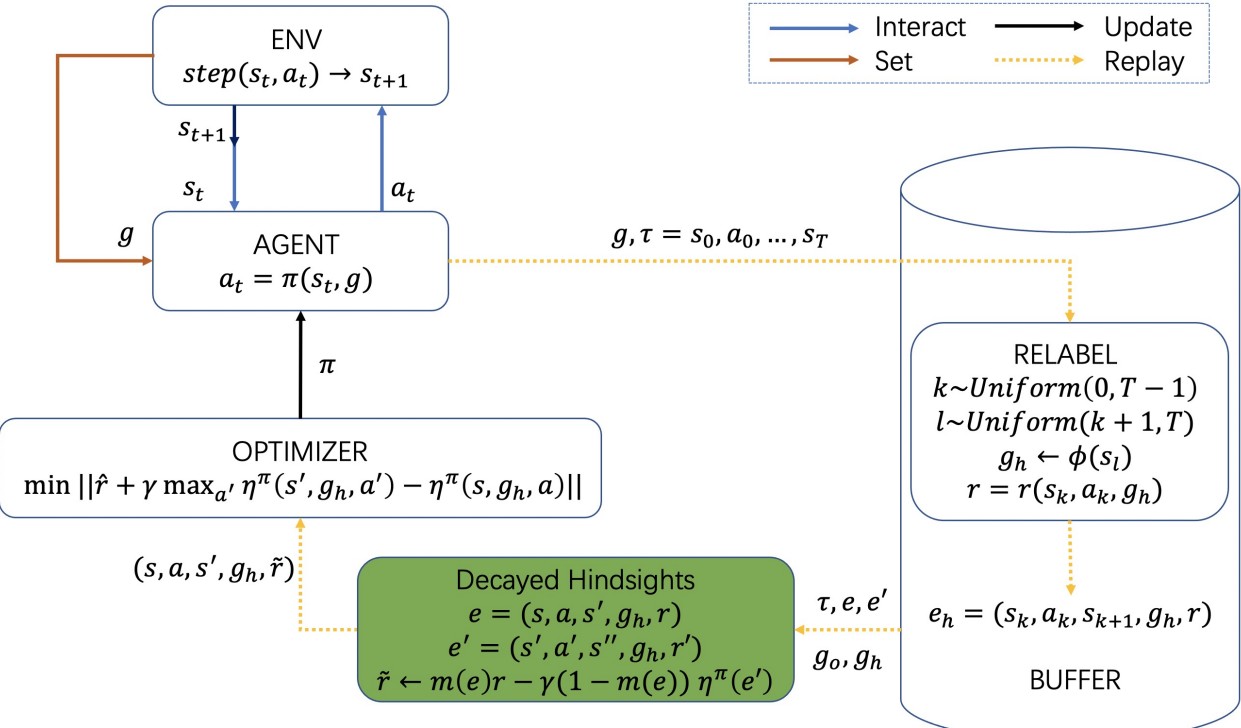

**Figure 3.** The overall framework for Decayed-HER (on the basis of the vanilla HER). It consists of collecting rollout data and replaying hindsight data. Our proposed DH enables optimized unbiased $\eta^\pi$ over hindsight experiences with modified hindsight rewards. If this DH module is disabled, it degrades to optimize $Q^\pi$ over hindsight experiences that are biased due to overestimated hindsights.

As shown in the framework, the training process runs along each module in a loop:

- Given the desired goal $g$ at the start of each episode, AGENT interacts with ENV with a goal-conditioned policy $\pi$ and stores the generated trajectory $\tau$ into BUFFER.
- Sample a trajectory from BUFFER and RELABEL experiences with hindsight goals and the resulting hindsight rewards. Here, we show a simple strategy for sampling replay goals: First, we randomly sample a state $s_k$ and a future state along the same trajectory after it, $s_l$. Then, we abstract the goal representation of $s_l$ as $g_h$ and store the relabeled $e_h = (s_k, a_k, s_{k+1}, g_h, r(s_k, a_k, g_h))$. If the decayed hindsights module is enabled, the hindsight rewards will be modified.
- OPTIMIZER samples experiences (that may be a mixed batch of original experiences and hindsight experiences) from BUFFER to update the policy $\pi$ of AGENT.

We apply DH to the decayed hindsights module, which is significantly different from other HER algorithms and is indispensable to the loop for unbiased objective $\eta^\pi$. Subsequently, we present the resulting Decayed-HER, which can incorporate decayed hindsights with any goal-conditioned off-policy RL algorithm, as shown in Algorithm 1.

---

**Algorithm 1** Decayed hindsight experience replay.

> **Input:** A goal-conditioned off-policy RL algorithm $\pi$, a strategy $\mathbb{S}$ for sampling replay goals, a reward function $r : \mathcal{S} \times \mathcal{A} \times \mathcal{G} \to \mathbb{R}$, max episodes num $M$, max interaction steps $T$ and policy update number $N$ of each episode

1   Initialize $\pi$, replay buffer $R$, constant $c = 0.99$, discounted factor $\gamma$

2   **for** $episode = 1 \to M$ **do**

3      // Set
     Sample a goal $g_o$ and an initial state $s_0$

5      // Interact
     Generate a $T-$steps trajectory $\tau = s_0, a_0, s_1, a_1, \dots, a_{T-1}, s_T$ using $\pi$

7      // Replay
     **for** $t = 0 \to T - 2$ **do**

8         $r_t := r(s_t, a_t, g_o)$

9         Store $e_o = (s_t, a_t, s_{t+1}, g_o, r_t)$ in $R$

10        Sample a set of additional goals $\mathcal{G} := \mathbb{S}(\tau)$

11        **for** $g_h \in \mathcal{G}$ **do**

12           Relabel $e = (s_t, a_t, s_{t+1}, g_h, r(s_t, a_t, g_h))$

13           Relabel $e' = (s_{t+1}, a_{t+1}, s_{t+2}, g_h, r(s_{t+1}, a_{t+1}, g_h))$

14           Approximate
           $m(e) \approx \min(exp\{\sum_i (\log \pi(a_i|s_i, g_h) - \log \pi(a_i|s_i, g_o))\}, c)$ using $\pi$

15           Decayed $\widetilde{r} = m(e)r(s_t, a_t, g_h) - \gamma(1 - m(e))\eta^\pi(s_{t+1}, g_h, a_{t+1})$

16           Store $e = (s_t, a_t, s_{t+1}, g_h, \widetilde{r})$ in $R$

17        **end**

18      **end**

19

20      // Update
     **for** $n = 1 \to N$ **do**

21        Sample a minibatch $B$ from the replay buffer $R$

22        Perform one step of optimization of $\pi$ using $B$

23      **end**

24   **end**

---

### 3.5. Offline Decayed-HER

While Algorithm 1 works for online multi-goal tasks directly, we consider the probability of applying it in offline settings, where it learns from a fixed dataset without further exploring the environment. Notice that there is no explicit goal in the locomotion tasks; we consider the final states of expert trajectories with high returns as goals.

The offline dataset can be regarded as a static replay buffer with trajectories pre-generated by diverse behavior policies. Without loss of generality, let $\mu : \mathcal{S} \times \mathcal{G} \to \mathcal{A}$ denote a universal behavior policy. In off-policy RL, the target policy $\pi$ can learn from experiences generated by any behavior policy as long as if $\mathbb{P}(a = \pi(s, g)) > 0$, we have $\mathbb{P}(a = \mu(s, g)) > 0$ for each $(s, g)$. Conversely, if any state-action pair is unavailable for any behavior policy, there will be an approximation error in the estimation of $Q^\pi$. One of the great challenges for offline RL is to address the problem of potentially learning from unavailable state-action pairs, i.e., out-of-distribution (OOD) data. Most existing offline RL algorithms adopt the conventional value-based framework but regularize the policy learning to match with the behavioral data to mitigate this problem. One stream of works [49–51] avoid OOD state–action value evaluation by constraining the target policy

within the distribution or support of the behavioral data. In general, these offline RL algorithms define the penalized value function as

$$V_D^\pi(s,g) = \sum_t \gamma^t \mathbb{E}_{s_t}[r(s_t, a_t, g) - \alpha D(\pi(a_t|s_t, g), \mu(a_t|s_t, g))]$$

where $D$ is a divergence function between distributions over actions (such as KL divergence), $\alpha > 0$ is the penalization factor. It requires a pre-estimated cloned policy $\mu$ since we do not have access to any $\mu$.

Based on the realization of $m(e)$, we provide a practical realization of $D$ in terms of a $T$-step trajectory $\tau$. Concretely, we take

$$\widetilde{r}(e) = m(e)r(e) - \gamma(1 - m(e))\hat{\eta}^\pi(e') - \alpha(1 - m(e))$$
$$= m(e)r(e) - (1 - m(e))(\gamma\hat{\eta}^\pi(e') + \alpha)$$

for hindsight experience $e$. The first term in $\widetilde{r}$ shows that we decrease the original $r$ with the dissimilarity. The second term represents the punishment proportional to $1 - m(e)$, which gets rid of the estimation of $\mu$.

## 4. Results

We evaluate Decayed-HER on goal-oriented tasks in both online and offline settings.

### 4.1. Online Multi-Goal Replay

#### 4.1.1. Experimental Settings

*HER variants.* Multi-goal replay with hindsights is the key role to enable goal-conditioned RL with sparse binary rewards. The framework is the same as the one for our proposed Decayed-HER, as shown in Figure 3. In Table 1, we give a detailed description of existing HER variants. The strategy $\mathbb{S}$ changes the distribution of replayed goals and introduces hindsight bias. The base variant HER [19] neglects the bias when uniformly replaying hindsight goals; ARCHER [34] variant assigns aggressive rewards to counter the bias by amplifying the hindsight rewards; BHER [35] counts the likelihood bias of regenerating sampled trajectories, then proposes an approximated batch projection to reduce the bias; Filtered-HER [52] rejects hindsight experience with goals already achieved to ease another kind of estimation bias. There are HER variants that prioritize hindsight goals: CHER [45], which evaluates hindsight goals and generates an implicit curriculum for progressive exploration, and MEP [23], which incorporates with HER and optimizes a weighted entropy-based objective for exploration. We compare Decayed-HER with other biased-reduced HER variants and show if DH can improve the performance of HER variants that neglect the hindsight bias.

**Table 1.** HER variants with different hindsight rewards. The vanilla HER neglects the bias when uniformly replaying hindsight goals. ARCHER, BHER, and our Decayed-HER try to address the bias on the modified hindsights. Filtered-HER filters meaningless experiences but makes no effect on hindsight bias. CHER and MEP variants pay more attention to how to sample hindsight goals for replay.

| | Reward for $e = (s, a, s', g_h)$ | Bias-Reduced | Hindsight Bias | Strategy $\mathbb{S}$ |
|---|---|---|---|---|
| HER [19] | $r$ | NO | − | |
| ARCHER [34] | $constant \cdot r$ | YES | YES | |
| Filtered-HER [52] | $r$ | YES | NO | random |
| BHER [35] | $e^{\sum_t(||a_t - \pi(s_t, g_o)||_2^2 - ||a_t - \pi(s_t, g_h)||_2^2)}r$ | YES | YES | |
| **Decayed-HER (ours)** | $m(e)r - \gamma(1 - m(e))\hat{\eta}^\pi(e')$ | YES | YES | |
| CHER [45] | $r$ | NO | − | curriculum-based |
| MEP [23] | $r$ | NO | − | entropy-regularized |

*Goal-oriented tasks.* We perform the evaluation on goal-oriented tasks based on Robotic Fetch and Hand simulators [22,36,53], as shown in Figure 1. In Table 2, we list the dimensions of states, goals, and actions for each task, as well as a brief description and the tolerance threshold. At the beginning of each episode, the agent obtains the desired goal. Once the agent approaches the desired goal within a tolerance threshold, we consider it a success. The comparison is mainly based on the averaged success rate on all the tasks.

**Table 2.** Goal-oriented continuous control tasks for evaluating HER variants.

|  | Task | Dim.(State) | Dim.(Goal) | Dim.(Action) | Description | Tolerance Threshold |
|---|---|---|---|---|---|---|
|  | Pushing | 10 | 3 | 4 | push and roll a box | 5 cm |
| Fetch | Sliding | 25 | 3 | 4 | hit a puck | 5 cm |
|  | Pick-and-Place | 25 | 3 | 4 | grasp and move a box | 5 cm |
|  | ManipulateBolck | 61 | 7 | 20 | rotate $x, y, z$ axes | 1 cm, 0.1 rad |
| Hand | ManipulateEgg | 61 | 7 | 20 | rotate $x, y, z$ axes | 1 cm, 0.1 rad |
|  | ManipulatePen | 61 | 7 | 20 | rotate $x, y$ axes | 5 cm, 0.1 rad |

*Controlling parameters.* In fairness, we implement Decayed-HER with a DDPG [7] algorithm exactly as the other state-of-the-art HER variants. All the variants share the common environment setting and the same experimental configuration (in Table 3). As for their private controlling parameters, we make a parameter tuning referring to the original work. We conduct experiments on a TITAN V with 8 Gpus and each test runs with 5 Cpu workers. For every update, parameters are averaged by 2 rollouts for each worker. More running works can improve the performance but increase the computation burden. We figured out that our experimental setting is capable of these tasks.

**Table 3.** Main controlling and training parameters for Decayed-HER.

| | |
|---|---|
| epoch | 50 |
| episode number $M$ | 50 |
| max steps $T$ | 50 for Fetch, 100 for Hand |
| buffer size $|R|$ | $10^6$ transitions |
| update number $N$ | 40 |
| batch size $|B|$ | 256 |

### 4.1.2. Results and Analysis

*Benchmark results.* After the policy update at each epoch, we perform 10 rollouts with the current policy and take the averaged success rate as its performance. We compare the performance of different bias-reduced HER variants and compare mean success rates in the final evaluation, which are averaged over 6 random seeds and all the workers. For each variant on every task, we list the final mean success rate and its ratio to the best-achieved results of all the variants. For ARCHER, we try different aggressive amplify factors and give the best result (labeled as ARCHER_1). Thus, we obtain the averaged evaluation results for each variant in Table 4.

As shown in Table 4, Decayed-HER obtains the best result on the multi-goal tasks with sparse rewards, which is 99.6% average performance on all the manipulation tasks. Concretely, it achieves the best performance on hand tasks while yielding competitive results on fetch tasks. ARCHER, which achieves the best average results on Fetch tasks, achieves a poor result on hand tasks. BHER, which only considers the regenerating bias of trajectories, also works well, especially on tasks ManipulateBolck and ManipulatePen. Except for bias-reduced correction on hindsight rewards, there is an additional operation of value projection in BHER [35], which transfers the bias correction from zero rewards to non-zero rewards. We will study the effect of this operation in future work.

**Table 4.** Final mean success rates comparison. We list the final average success rate for each variant on each task and its ratio to the best achieved average success rate on the task.

| Variant<br>Task | HER | ARCHER_1 | Filtered-HER | BHER | Decayed-HER (Ours) |
|---|---|---|---|---|---|
| Pushing | 0.996/99.9% | 0.994/99.7% | **0.997/100.0%** | 0.995/99.8% | 0.996/99.9% |
| Sliding | 0.549/86.9% | **0.632/100.0%** | 0.554/87.7% | 0.613/97.0% | 0.628/99.4% |
| Pick-and-Place | 0.811/87.0% | **0.932/100.0%** | 0.897/96.2% | 0.864/92.7% | 0.918/98.5% |
| ManipulateBolck | 0.454/81.1% | 0.189/33.8% | 0.444/79.3% | 0.51/91.1% | **0.56/100.0%** |
| ManipulateEgg | 0.136/78.2% | 0.011/6.3% | 0.119/68.4% | 0.113/64.9% | **0.174/100.0%** |
| ManipulatePen | 0.173/74.6% | 0.146/62.9% | 0.171/73.7% | 0.186/80.2% | **0.232/100.0%** |
| Average success rate | 0.520 | 0.484 | 0.530 | 0.547 | **0.585** |
| Average relative ratio | 84.6% | 67.1% | 84.2% | 87.6% | **99.6%** |

By comparing to HER and Filtered-HER, we show that Decayed-HER achieves better final performances after adopting Decayed Hindsights. By comparing to ARCHER, we show that the adaptable decayed rewards work better than simply assigning constant-factor amplified rewards. By comparing to BHER that neglects the skewed values, we show that the proposed surrogate objective is effective in value estimation and achieving better results. In Figure 4, we additionally provide the learning curves for all the results on the pushing and ManipulateEgg tasks to show the stable learning process and high sample efficiency of Decayed-HER.

*Learning curves.* In Figure 4, we plot learning curves of different bias-reduced HER variants, which show the median test success rate with shaded areas representing performance fluctuations along with the training process. For ARCHER, we plot the best one (labeled as ARCHER_1) and the average one (labeled as ARCHER_2) after trying different constant aggressive amplify factors.

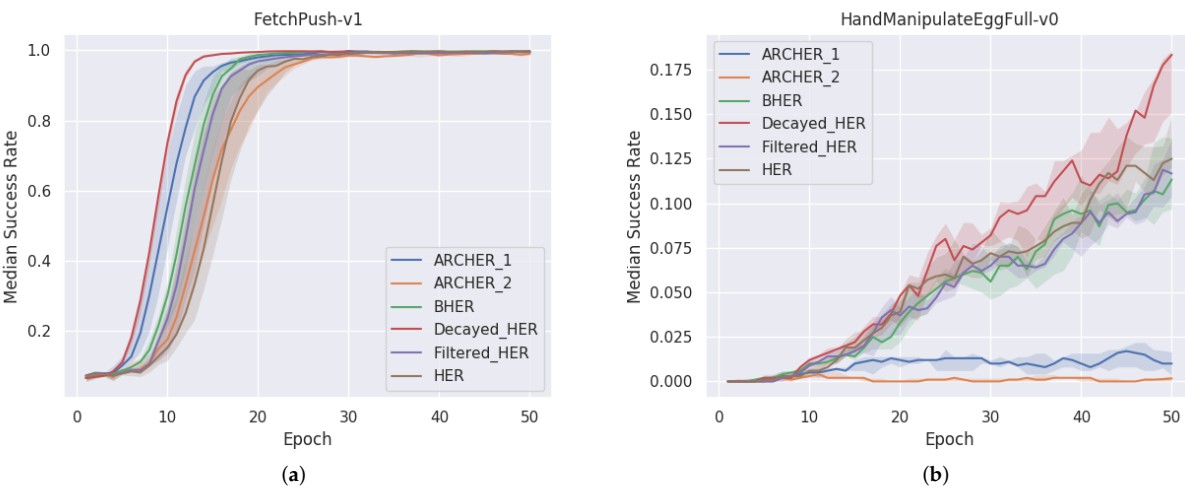

**Figure 4.** Learning curves for bias-reduced HER variants on (**a**) Pushing, (**b**) ManipulateEgg. The bold line shows the median test success rate over 6 random seeds and the shaded areas represent the 25th to 75th percentile. For ARCHER, we plot the best and averaged results with two different aggressive amplify factors. Our method achieves the best average performance both on sample efficiency and final median success rate.

We can see that (1) when achieving some success rate, our Decayed-HER uses fewer epochs most of the time. Thus it obtains the best sample efficiency and achieves the highest success rate on task Pushing and ManipulateEgg. (2) ARCHER is fragile on hand tasks, which indicates that amplifying the hindsight rewards by a constant may be dangerous for complex manipulation tasks. (3) While HER realizes stable baseline results, our DH successfully improves HER in both the sample efficiency and final success rate.

*Ablation study.* We perform ablation studies on decayed Hindsights by applying it to state-of-the-art non-bias-reduced HER variants, i.e., the CHER variant and MEP variant. These variants significantly encourage exploration across the goal space, which makes them good testbeds for evaluating the effect of DH assigned to different goals. We conduct experiments on the pick-and-place task and sliding task in the fetch environment across three random seeds. The variants with decayed hindsight are labeled as CHER_DH and MEP_DH. Learning curves in Figure 5 show that with the effect of DH, the performance of the original variant has an obvious improvement.

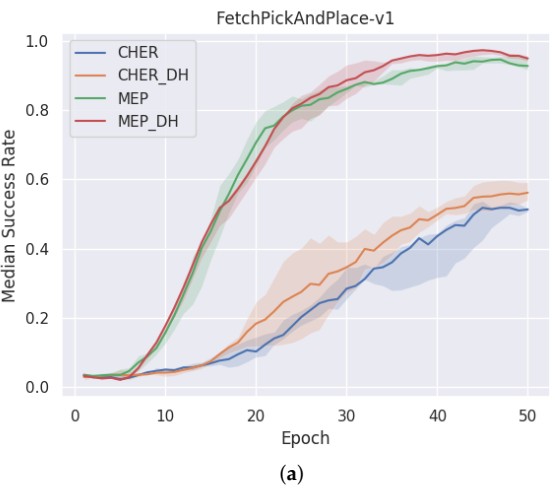
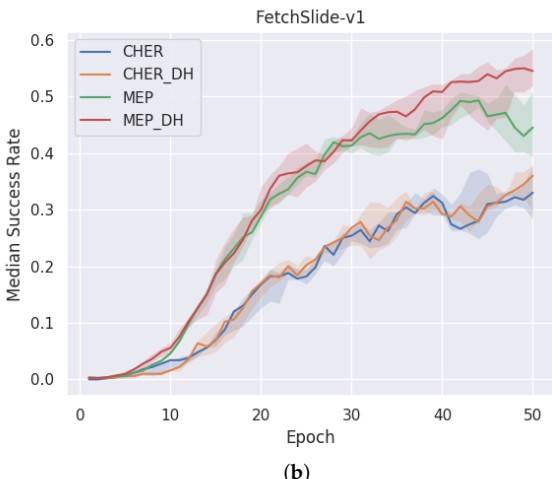

**Figure 5.** Ablation study on DH by performing experiments on (**a**) Pick-and-Place; (**b**) Sliding. Variants with DH perform better or no worse than before.

### 4.2. Offline Decayed-HER

#### 4.2.1. Experimental Settings

*Offline RL as goal-conditioned RL.* We consider the offline dataset D4RL [54], which contains different sub-sets generated by different strategies and provides a popular benchmark of offline RL. In Table 5, we list the sub-set information for continuous locomotion tasks halfcheetah, walker2d and hopper (shown in Figure 1). In the first place, it stops the training of a policy early and then constructs the subsets: the 'medium-replay' sub-set contains all the experiences during the training; the 'medium' sub-set collects random rollouts with the partially-trained policy; and the 'medium-expert' sub-set mixes expert demonstrations with sub-optimal data, which are generated via the partially trained policy or by unrolling a uniform-at-random policy. With more sub-optimal data, behavior-regularized methods will obtain a more poor result.

The sub-sets are not generated via goal-conditioned policies. To solve offline RL as goal-conditioned RL, we adopt a goal-conditioned policy and random sample states with high accumulated rewards as goals to guide the training. It in a way enriches training samples. When evaluating the performance, RL agent sets the sampled states as goals and then rollouts. Once the agent approaches the states within a tolerance threshold (i.e., the distance of normalized states is small than a pre-defined constant), we consider it a success. Except for such sparse goal-reaching signals, the locomotion tasks receive external rewards. D4RL transforms accumulated rewards into normalized scores compared to its baseline. We adopt the setting and compare the average normalized scores on all the tasks.

**Table 5.** Offline continuous locomotion tasks in D4RL.

| Task | Sub-Set | Dim.(State) | Dim.(Action) | Size | Description |
|---|---|---|---|---|---|
| halfcheetah | medium | 17 | 6 | $10^6$ | generated by a partially-trained policy |
| | medium-replay | 17 | 6 | 101,000 | all samples in the replay buffer |
| | medium-expert | 17 | 6 | $2 \times 10^6$ | mixed expert and sub-optimal data |
| walker2d | medium | 17 | 3 | $10^6$ | generated by a partially-trained policy |
| | medium-replay | 17 | 3 | 100,930 | all samples in the replay buffer |
| | medium-expert | 17 | 3 | $2 \times 10^6$ | mixed expert and sub-optimal data |
| hopper | medium | 11 | 3 | $10^6$ | generated by a partially-trained policy |
| | medium-replay | 11 | 3 | 200,920 | all samples in the replay buffer |
| | medium-expert | 11 | 3 | $2 \times 10^6$ | mixed expert and sub-optimal data |

*Controlling parameters.* To give a fair comparison, we implement offline Decayed-HER with the vanilla BRAC [51]. They share a common environment setting and the same experimental configuration. In Table 6, we list the main parameters for training and evaluation. The performance is averaged over 100 rollouts at each evaluation.

**Table 6.** Main controlling parameters for offline Decayed-HER.

| | |
|---|---|
| training steps | $10^6$ |
| batch size | 256 |
| goals selection | random, within high-rewarding trajectories |
| number of rollouts | 100 |
| batch size | 256 |
| max steps | 1000 |

### 4.2.2. Results and Analysis

*Benchmark results.* We compare our offline Decayed-HER with value-based behavior-regularized offline RL: BCQ [49], BEAR [50], BRAC [51], although they are not initially designed for goal-conditioned learning. In Table 7, we can see that (1) our method shows competitive results on the locomotion tasks, which indicates the original tasks can be solved as a goal-conditioned RL; (2) in almost all the sub-sets, our method outperforms the vanilla BRAC, which shows the effect of decayed hindsights; (3) our method achieves the best average normalized scores on 'medium' sub-sets for all the tasks, which lies in the fact that in such sub-sets, our method is able to well estimate the static distributions.

The proposed offline Decayed-HER is based on BRAC but works better than BRAC, which indicates the effectiveness of applying DH to the offline settings. By comparing it to BCQ and BEAR, we show that offline Decayed-HER achieves competitive performance with other value-based offline algorithms, especially in 'medium' datasets generated by a stable policy.

*Learning curves.* Except for the comparison on the final performance, we compare the performance of goal-conditioned RL with and without DH on the task hopper, and plot learning curves in Figure 6. It shows the average normalized scores along with the training process over each sub-set. We can see that DH substantially improves the performance of the basic goal-conditioned policy on 'medium' and 'medium-expert' sub-sets. In the 'medium-replay' sub-set, the one with DH yields a fluctuated average performance with fewer samples.

**Table 7.** Average normalized scores on D4RL benchmark tasks. The evaluations are averaged on 100 random seeds and normalized by following [54]. Our method shows promising results on the locomotion tasks.

| Method Task | BCQ | BEAR | BRAC | Offline Decayed-HER (Ours) |
|---|---|---|---|---|
| halfcheetah-medium | 47.0 | 41.7 | 46.3 | **48.1** |
| halfcheetah-medium-replay | 40.4 | 38.6 | **47.7** | 45.2 |
| halfcheetah-medium-expert | **89.1** | 53.4 | 41.9 | 86.7 |
| walker2d-medium | 72.6 | 59.1 | 81.1 | **82.7** |
| walker2d-medium-replay | **52.1** | 19.2 | 0.9 | 36.4 |
| walker2d-medium-expert | **109.5** | 40.1 | 81.6 | 108.4 |
| hopper-medium | 56.7 | 52.1 | 31.1 | **71.2** |
| hopper-medium-replay | 53.3 | 33.7 | 0.6 | **63.5** |
| hopper-medium-expert | 81.8 | **96.3** | 0.8 | 68.2 |

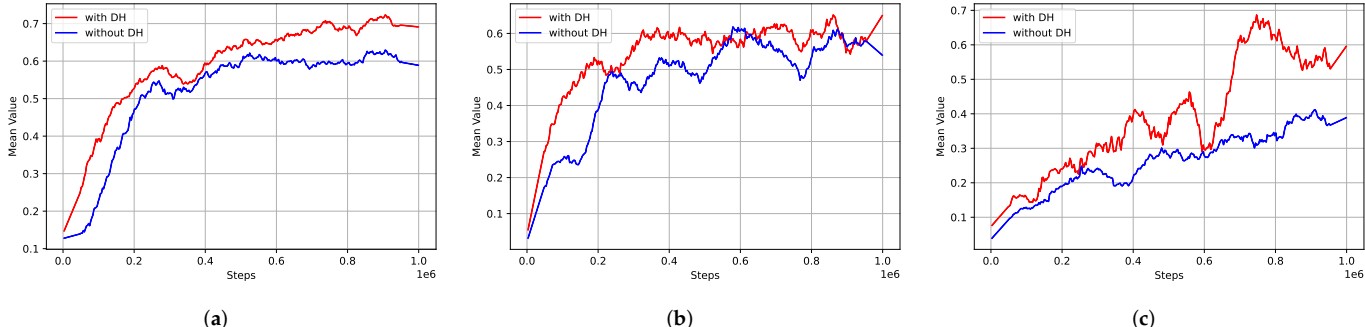

(**a**)          (**b**)          (**c**)

**Figure 6.** Ablation study on DH over datasets (**a**) medium, (**b**) medium-replay and (**c**) medium-expert. Offline goal-conditioned RL with DH performs better than before.

## 5. Discussion and Conclusions

In recent years, the rapid development of neural networks has made it possible to efficiently and continuously control high-dimensional tasks. The approximation errors introduced by the neural networks are from different sources. In value-based RL, function approximation errors lead to overestimated value estimates and sub-optimal policies [47]. In our work, we focus on another approximation error from the abuse of hindsights, which widely exist in HER [19]. While existing works on hindsight bias neglect the effect of distribution shift on rewards, we instead derive a surrogate objective with modified rewards, i.e., decayed hindsights.

In this paper, we propose decayed hindsights to ease the influence of the overestimation of regenerating the hindsight experience. We analyze the hindsight bias in vanilla HER and approximate an unbiased MDP with decayed rewards. It leads to an efficient, bias-reduced consistent multi-goal hindsight replay, namely Decayed-HER. With its help, continuous control with sparse rewards in high dimensions can be consistently and effectively solved via multi-goal RL. In general, we incorporate our decayed Hindsights both with online multi-goal replay and offline behavior-regularized policies. Experiments show that it enables obtaining high performance and stabilizing the learning from sparse rewards. By framing typical offline RL into goal-conditioned RL, we are able to apply goal-conditioned policies in solving tasks without explicit goals.

In future work, we will further explore the hindsight bias in the goal-conditioned self-imitation method. It adopts a similar idea to hindsight replay but can be realized in a policy-based manner [40–43]. The hindsight bias, which reflects the difference between the estimated value and actual return, may lead to unstable and uncontrollable policy updates. It is essential to directly counter the bias in addition to trust the region policy optimization [9,55].

**Funding:** This research received no external funding.

**Institutional Review Board Statement:** Not applicable.

**Informed Consent Statement:** Not applicable.

**Data Availability Statement:** Not applicable.

**Conflicts of Interest:** The author declares no conflict of interest.

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
