# Peer review of "Consistent Experience Replay in High-Dimensional Continuous Control with Decayed Hindsights"

_machines, doi:10.3390/machines10100856_

Round 1

Reviewer 1 Report

This paper is an interesting study for Continuous Control.

The paper lacks novelty. However, The outstanding results and content deserve to be introduced to the readers.

There are two point for improving the paper.

1. While the results of this paper are impressive, the method does not feel significant.

I ask an author to share their source code and experimental results.

If you use OpenAI Gym for sharing, more readers will read and utilize this paper.

2. The paper's English skill is rough.

In particular, there are so many sentences that start with To, it feels a bit unprofessional for English-speaking authors.

Reviewer 2 Report

Proposed manuscript deals an extremely interesting topic of machine learning in complex robotics. Machne (reinforcement) learning is a crucial part of modern robot control therefore I can see a grat potential in the topic. Author describe describe novel approach based  on hindsight experience replay.

I am little bit confused because the manuscript is carefully written but it is somewhere between theoretical and applied research and I can see the biggest shortcoming in the missing experiments.

So I would like to advise the authors to slightly rebuilt the manuscript, add verification experiments and prove the approach works properly.

Owing facts mentioned above I recommend to accept the manuscript after minor revision.

Reviewer 3 Report

The manipulation of complex robotics, which is in general high dimensional continuous control without an accurate dynamic model, summons studies and applications of reinforcement learning (RL) algorithms. Typically, RL learns with the objective of maximizing the accumulated rewards from interactions with the environment. In reality, external rewards are not trivial, which depend on either expert knowledge or domain priors. Recent advances on hindsight experience replay (HER) instead enable a robotic to learn from the automatically generated sparse and binary rewards indicating whether it reaches the desired goals or pseudo goals. However, HER inevitably introduces hindsight bias that skews the optimal control, since the replays against the achieved pseudo goals may often differ from the exploration for the desired goals. To tackle the problem, the authors analyzed the skewed objective and induce the decayed hindsights (DH), which enables consistent multi-goal experience replay via countering the bias between exploration and hindsight replay. They implement DH for goal-conditioned RL both in online and offline settings. Experiments on online robotic control tasks demonstrate that DH achieves the best average performance and is competitive with state-of-the-art replay strategies. Experiments on offline robotic control
tasks show that DH substantially improves the ability to extract near-optimal policies from offline datasets. A couple of references from this journal should be added in reference section. The following references should be added in reference section:

1. M.K. Sharma, Kamini, N. Dhiman, V.N. Mishra, H.G. Rosales, A. Dhaka, A. Nandal, E.G. Fernández, T.A. Ramirez, L.N. Mishra, A Fuzzy Optimization Technique for Multi-Objective Aspirational Level Fractional Transportation Problem, Symmetry, Vol. 13, No. 8, (2021), Article ID: 1465. DOI: https://doi.org/10.3390/sym13081465

2. M.K. Sharma, Sadhna, A.K. Bhargava, S. Kumar, L. Rathour, L.N. Mishra, S. Pandey, A FERMATEAN FUZZY RANKING FUNCTION IN OPTIMIZATION OF INTUITIONISTIC FUZZY TRANSPORTATION PROBLEMS, Advanced Mathematical Models & Applications, Vol. 7, No. 2, (2022), 191-204.

Recommendation: Based on above report, manuscript is accepted in this journal after minor revision.

Author Response

Point 1:A couple of references from this journal should be added in the reference section. 

Response 1:We found the references recommended in the review report are a bit far from the topic of the manuscript. Concretely, the references aim at fuzzy optimization and multi-objective tasks, whilst this manuscript focuses on multi-goal reinforcement learning. Therefore, we do not cite the recommended references but add a couple of relevant references:

  1. Prianto, E.; Park, J.-H.; Bae, J.-H.; Kim, J.-S. Deep Reinforcement Learning-Based Path Planning for Multi-Arm Manipulators with Periodically Moving Obstacles. Appl. Sci. 202111, 2587. https://doi.org/10.3390/app11062587
  2. Prianto, E.; Kim, M.; Park, J.-H.; Bae, J.-H.; Kim, J.-S. Path Planning for Multi-Arm Manipulators Using Deep Reinforcement Learning: Soft Actor–Critic with Hindsight Experience Replay. Sensors 202020, 5911. https://doi.org/10.3390/s20205911
  3. Kim, M.; Han, D.-K.; Park, J.-H.; Kim, J.-S. Motion Planning of Robot Manipulators for a Smoother Path Using a Twin Delayed Deep Deterministic Policy Gradient with Hindsight Experience Replay. Appl. Sci. 202010, 575. https://doi.org/10.3390/app10020575
  4. Pendleton, S.D.; Andersen, H.; Du, X.; Shen, X.; Meghjani, M.; Eng, Y.H.; Rus, D.; Ang, M.H. Perception, Planning, Control, and Coordination for Autonomous Vehicles. Machines 2017, 5, 6. https://doi.org/10.3390/machines5010006

Besides, we checked the grammar and spelling of the manuscript as carefully as possible. We would like to thank the reviewer again for taking the time to review our manuscript.